# Association between Young People’s Neighbourhoods’ Characteristics and Health Risk Factors in Saudi Arabia

**DOI:** 10.3390/healthcare12111120

**Published:** 2024-05-30

**Authors:** Anwar Al-Nuaim, Abdulmalek K. Bursais, Marwa M. Hassan, Abdulrahman I. Alaqil, Peter Collins, Ayazullah Safi

**Affiliations:** 1Physical Education Department, Education College, King Faisal University, Al-Ahsa 31982, Saudi Arabiaaialaqil@kfu.edu.sa (A.I.A.); 2Faculty of Education Health and Wellbeing, University of Wolverhampton, Wolverhampton WV1 1LY, UK; 3Department of Public Health, Centre for Life and Sport Science (C-LaSS), Birmingham City University, Birmingham B15 3TN, UK

**Keywords:** neighbourhood, characteristics, health, risk factors, youth, Saudi, physical activity and BMI, geographical location, cultural attitudes and social environment

## Abstract

Introduction: A neighbourhood’s environmental characteristics can positively or negatively influence health and well-being. To date, no studies have examined this concept in the context of Saudi Arabian youth. Therefore, this study aimed to evaluate the association between a neighbourhood’s environmental characteristics and health risk factors among Saudi Arabian youth. Methods: A total of 335 secondary-school students (175 males, 160 females), aged 15–19 years old, participated. Body mass index (BMI) and waist circumference measurements were taken, and physical activity (steps) was measured via pedometer. The perceived neighbourhood environment was assessed using the International Physical Activity Questionnaire Environment Module (IPAQ-E). Results: Significant differences were found between the youths from urban, rural farm, and rural desert locations in terms of BMI, waist circumference, daily steps, accessibility, infrastructure, social environment, household vehicles, safety, and access to facilities (*p* < 0.001). Rural desert youths were less active, and males (26.43 + 8.13) and females (24.68 + 5.03) had higher BMIs compared to the youths from other areas. Chi-square analysis revealed a significant difference (*χ*^2^_1_ = 12.664, *p* < 0.001) between the genders as to social-environment perceptions. Males perceived their neighbourhood as a social environment more than was reported by females (68.39% and 50.28%, respectively). Pearson’s correlation revealed negative significant relationships between steps and both safety of neighbourhood (r = −0.235, *p* < 0.001) and crime rate (r = −0.281, *p* < 0.001). Discussion: Geographical location, cultural attitudes, lack of facilities, and accessibility impact youth physical-activity engagement and weight status; this includes environmental variables such as residential density, neighbourhood safety, household motor vehicles, and social environment. Conclusions: This is the first study examining associations with neighbourhood environments in the youths of the Kingdom of Saudi Arabia. Significant associations and geographical differences were found. More research and policy interventions to address neighbourhoods’ environmental characteristics and health risk factors relative to Saudi Arabian youth are warranted.

## 1. Introduction

The influences of environmental characteristics, including nature, air quality, temperature, and cleanliness are expected to have long-term, significant positive or negative impacts on the health and well-being of the population [1,2,3]. Previous research has indicated that the impacts of a neighbourhood’s environmental characteristics, including residential density, accessibility, recreational facilities, aesthetics, and safety, play a major role in influencing physical activity (PA), health, and well-being [4,5,6]. PA is defined as any bodily movements produced by the skeletal muscles, and resulting in energy expenditure [7]. Previous research has suggested that if PA engagement is supported environmentally, socially, and culturally, people will perceive it as being vital for health, and are likely to face fewer barriers to engaging in PA and leading a healthy and active lifestyle, compared to people in countries where PA is not a social norm [8]. The benefits of engaging in regular PA are well-known, including reducing the risk of mortality and the incidence of cardiovascular diseases, diabetes, and some forms of cancers [9,10,11,12]. Despite the well-known benefits associated with PA engagement, research shows that a large proportion of the population around the world is failing to partake of the recommended PA guideline of 150 min of moderate-to-vigorous physical activity (MVPA) for adults each week, or the minimum of 60 min of MVPA per day for children and young people [13]. Lack of time, workload, the expansion of urbanisation, different modes of transportation, availability of entertainment gadgets, culture, autonomy, and lack of knowledge are some of the reported barriers to PA participation and incentives to an unhealthy lifestyle across the board [8,14,15,16,17]. 

As previous research has reported, the prevalence of physical inactivity among the population of the Kingdom Saudi Arabia (KSA) is high, as compared to Western countries including the UK and the USA [18]. Moreover, recent research has reported that over 90% of the KSA population leads a sedentary lifestyle; this is particularly prevalent among females [16,17]. Some of the key factors contributing to the KSA’s population’s sedentary lifestyle include neighbourhood characteristics, including the rapid development of the country and the enhancement and availability of cars/transport and technological gadgets [17]. A sedentary lifestyle contributes to morbidity and mortality; thus, the exposure of young people to an environment that supports PA participation as part of a healthy and active lifestyle is paramount. For instance, previous research has focused on individual demographics in seeking to promote PA and prevent obesity, and the results suggested that the influence of environmental characteristics was expected to have positive impact on the population [19,20]. As previously mentioned, people who regard PA as being important for health and well-being are more likely to engage in PA and lead a healthy and active lifestyle, compared to people in countries where PA is not regarded as essential [8]. Furthermore, comparison was made of the barriers to PA for Afghans in the UK and those living in Afghanistan, and the findings suggested that Afghan females living in the UK felt that some aspects of their daily life were more of a barrier to PA, compared to their counterparts living in Afghanistan, and this is despite the UK having much more PA-supportive infrastructure and resources. Also, barriers such as lack of single-sex facilities, not being able to participate in PA with males, and having to be fully covered outside of the home were identified as important by all Afghan females, irrespective of where they resided, but these were more of a barrier for UK-based females [8]. 

Previous research has also demonstrated that cultural attitudes impact PA engagement and healthy lifestyle, as evidenced in a study that compared youth from the UK with their counterparts in Saudi Arabia [8,18]. Previous research has indicated that environmental characteristics, including access to amenities and recreational facilities and safety could influence PA participation, obesity levels, and incidence of a healthy lifestyle. For instance, women living in urban areas reported having higher mean values for waist and hip circumferences and skinfold thicknesses, as well as for lean body mass and total fat, compared to those living in the rural areas of Kaur and Talwar [4,20,21]. Comparative results show that people from rural areas were reported as engaging in larger amounts of PA compared to their counterparts living in urban areas [22,23]. Previous research reported that the availability and accessibility of facilities in close proximity, including parks, were linked to higher PA engagement, especially in girls, whereas close proximity to retail stores was associated with boys’ PA engagement [24]. Furthermore, research suggested that participants living within approximately one kilometre of a park were found to be five times more likely to be considered to have a healthy weight than were people without nearby parks or playgrounds [25,26]. Larson et al. suggested that easy access to supermarkets and limited access to convenience stores plays a positive role in accessing healthier nutrition and a lower level of obesity amongst residents [25]. Dengel et al. found that body fat and metabolic syndrome were linked to increased distance to fast food and convenience stores [27]. 

Moreover, it has been suggested that the safety of the local community and the wider society play vital roles in people’s PA and healthy behaviour [28,29,30,31,32]. On the other hand, the total number of incidents of serious crime in adolescents’ neighbourhoods was linked to decreased PA engagement [31,32,33] and increases in BMI [34,35]. Studies previously conducted focused on the impact of the built environment on PA and health. However, most of the existing research was conducted in developed countries, especially in the Western world, mainly in the USA and UK [1,36,37,38]. Studies focusing on neighbourhood characteristics and their impacts on health among youth in Saudi Arabia are limited in number. Therefore, this study aimed to assess the correlation between a neighbourhood’s environmental characteristics and health risk factors among KSA youth.

## 2. Methods of Investigation

### 2.1. Participants and Procedures

The data collection commenced after gaining institutional ethical approval. The participants were invited from across the Al-Ahsa Governorate region, including from areas such as urban, rural, and rural desert and rural farm, to take part in this study. The participants were recruited from public and private secondary schools; these included boys- and girls-only schools. A total of four schools for each gender were selected to acquire a representation of the different geographical locations, including urban, rural desert, and rural farm. The sample size was calculated to assess the ratio of residents within a border error of ±0.05 and a 95% confidence level. The total sample consisted of (*n* = 335) secondary school students, both boys (*n* = 175) and girls (*n* = 160).

### 2.2. BMI Measurement

Body weight was measured to the nearest 100 g using Seca weight digital scales (Seca Ltd., Hamburg, Germany). Standing height was measured to the nearest 0.5 centimetre using a Seca portable height measure (Seca Ltd., Hamburg, Germany). BMI was calculated using the formula: weight (kg)/height (m^2^). Sex- and age-specific International Obesity Task Force (IOTF) BMI cut-offs were used to define normal weight, overweight, and obesity [39]. For instance, participants were classified as follows: BMI < 5th percentile = “underweight”, BMI ≥ 5th percentile and <85th percentile = “normal weight”, BMI ≥ 85th and ≤94.9th percentile = “overweight”, and BMI ≥ 95th percentile = “obese” [40].

### 2.3. Waist Circumference Measurement

In this study the Pollock (1990) guideline was applied for the waist circumference measurement, which was collected to the nearest 0.5 cm. The waist circumference cut-off point was set at a waist-to-height ratio of 0.5 [41]. Participants whose waist circumference measurement was over 0.5 cm were considered to be ‘at risk’ of cardiovascular diseases. Previous research has noted that waist circumference is an easy form of measurement that can provide an indication of fat distribution in people, regardless of gender, race, and adiposity [41,42]. 

### 2.4. Physical Activity (Steps) Measures

The piezoelectric (Digiwalker) pedometer, New-Lifestyles NL-2000 Activity Monitor (NEW-LIFESTYLES, Inc., Less Summit, MO, USA), was used to objectively measure PA (steps) engagement. Previous research has supported the use of a pedometer, which measures PA levels objectively and reports valid and reliable data [43,44,45,46,47]. In this study, prior to handing out the piezoelectric devices to the participants, training was provided during their physical education class. Participants were instructed about the method of attachment of the piezoelectric around the waist, asked not to wear the piezoelectric when sleeping or during any water activities such as bathing or swimming, and to put it on before leaving for school. The piezoelectric was given to participants in schools on Wednesday and collected on Sunday morning (Sunday is the start of the week in Saudi Arabia). The collection of steps via pedometer was in line with previous research that assessed PA/steps for a four-day period, including two weekdays and a weekend [47,48,49]. Pedometer values were taken as the average number of steps per day weighted according to the ratio of days. In this study, participants were divided into two categories, active and inactive, across gender. For instance the cut-off point for the active category for females was 11,000 steps, and for boys it was set to 13,000 steps per day; anything less than the outlined numbers of steps was considered to be inactive [50]. 

### 2.5. Environmental Measurement

The International Physical Activity Questionnaire Environment Module (IPAQ-E) was used to measure perceived neighbourhood environmental characteristics linked to PA [51,52]. The IPAQ-E consists of 17 questions: seven core items, four recommended items, and six optional items. For the purpose of this study, items on the IPAQ E- module were classified into eight categories [52,53]: residential density (one item), access to destinations (three items), neighbourhood infrastructure (four items), aesthetic qualities (one item), recreational facilities (one item), social environment (one item), household motor vehicle (one item), and neighbourhood safety (four items). These categories have been shown to demonstrate moderate reliability coefficients in previous studies [53,54]. Two questions required participants to select from a list of options: these related to their housing type and the number of cars in their household. The other 15 items were based on a 4-point Likert scale ranging from ‘strongly disagree’ to ‘strongly agree’ (as well as ‘don’t know’ or ‘doesn’t apply’ options), with participants required to answer each statement that applied to them, as described in a previous study [52].

## 3. Statistical Analysis

A range of statistical tests were conducted to understand the correlations and differences among participants’ health and lifestyle habits across locations and gender. The descriptive characteristics, including means and standard deviations for the main dependent continuous variables of the sample, were determined as outlined in Table 1. In addition, comparisons were drawn between the lifestyle habits of youth across a range of activities, as well as gender, geographical location, and age group; analyses were conducted using 2-way and 3-way analyses of variance (ANOVA) on participants PA levels, BMI, and abdominal circumferences. The Bonferroni post-hoc test and Chi-squared analyses, as well as Pearson’s correlations, were used to analyse the data across gender and as to different locations. The statistical significance for analysis was set to *p <* 0.05 and 95% confidence intervals, using version 25 in SPSS.

## 4. Results

### 4.1. Differences in Neighbourhoods’ Characteristics

#### 4.1.1. Residential Density

The only question that assessed residential density was as to the main type of housing in the neighbourhood. Many males and females lived in houses (86.36% and 89.39%, respectively). Chi-square analysis revealed significant differences as to a youth’s accommodation between different geographical locations (*χ*^2^_2_ = 8.580, *p* = 0.014). The highest percentage of youths living in houses was found among youths from the rural desert (95.79%), whereas the lowest percentage was found among youths from rural farms (82.52%), while 86.59% of urban youths lived in houses. The univariate ANOVA showed no significant differences between types of accommodation as to average steps per day, BMI, or abdominal circumference (*p* > 0.05).

#### 4.1.2. Access to Destination

There were three items assessed under the ‘access to destination’ category, as outlined in Table 2. In general, 53.16% of youths perceived a lack of accessibility in their neighbourhood (i.e., shops, supermarkets, places, transportation, etc.). The lack of accessibility was obviously tested in the proposition “there are many shops within walking distance of the house”. In response to this question, 92.89% of participants responded as strongly disagreeing or somewhat disagreeing. Chi-square analysis revealed a significant difference between the perceptions of youths from different geographical locations (*χ*^2^_2_ = 40.727, *p* < 0.05). Youths from urban areas perceived more accessibility in their neighbourhood (75.90%) than did youths from rural farm and rural desert areas (34.57% and 53.93%, respectively). There was no significant difference between youth’s perception of accessibility as to average steps per day, BMI and abdominal circumference.

#### 4.1.3. Neighbourhood Infrastructure

Four items pertained to neighbourhood infrastructures; as outlined in Table 2, 29.55% of participants perceived either a lack of (not having) or poorly maintained sidewalks and bicycle facilities in their neighbourhood. Univariate ANOVA showed a significant difference between youths as to their perceptions of the neighbourhood infrastructure. Rural desert youths significantly disagreed more with the proposition that the neighbourhood had good infrastructure, as compared to urban youths (*p* < 0.001) and rural farm youths (*p* = 0.008). Moreover, Chi-square analysis indicated a significant difference between the perceptions of youths from different geographical locations (*χ*^2^_2_ = 15.743, *p* < 0.001): 33.55% of urban youths perceived their neighbourhood as having good infrastructure, whereas 18.75% of rural farm youths and 11.39% of rural desert youths reported that their neighbourhoods had good infrastructure. There was no significant difference between youths’ perception of neighbourhood infrastructure as to average steps per day, BMI, and abdominal circumference. 

#### 4.1.4. Aesthetic Qualities

The only question on aesthetic quality tested whether “there are many interesting things to look at while walking in my neighbourhood”. Chi-square analysis revealed a significant difference (*χ*^2^_1_ = 8.131, *p* = 0.003) between genders in perceptions of aesthetic qualities; 77.04% of males either strongly disagreed or somewhat disagreed, whereas the percentage of females for the same was 88.27%. Regarding geographical location differences, Chi-square analysis revealed a highly significant difference between youths’ responses (*χ*^2^_2_ = 15.537, *p* < 0.001): 25.70% of urban youths perceived aesthetic qualities in their neighbourhood, whereas 10.78% of rural farm youths and 9.57% of rural desert youths perceived aesthetic qualities in their neighbourhood. There was no significant difference between youths’ perception of aesthetic qualities as to average steps per day and/or weight status (*p* > 0.05). 

#### 4.1.5. Social Environment

The Chi-square analysis revealed a significant difference (*χ*^2^_1_ = 12.664, *p* < 0.001) between genders as to social-environment perceptions. For instance, males perceived their neighbourhood as a social environment more than reported by females (68.39% and 50.28%, respectively). In terms of geographical location, Chi-square analysis revealed a highly significant difference between youths’ responses. Most of the rural farm youths (70.30%) perceived their neighbourhood as a social environment, whereas the percentages for urban and rural desert youths were 62.15% and 43.62%, respectively. Univariate ANOVA revealed a significant difference (*p* = 0.017) between youths’ perceptions of their neighbourhoods as a social environment as to total PA engagement. Youths who perceived more of a social environment in their neighbourhood were more significantly active than youths who perceived less of a social environment (*p* < 0.05). However, Pearson’s correlation revealed no significant relationship between steps and social environment (*r* = 0.077, *p* = 0.164; as outlined in Table 3). However, there was no significant difference found in social environment perception as to BMI and waist circumference (*p* > 0.050). 

#### 4.1.6. Household Motor Vehicles

The only question addressing the social environment was “how many motor vehicles in working order”. Univariate ANOVA revealed a significant difference between geographical locations as to household motor vehicles (*p* = 0.019). Rural desert youths had significantly more household motor vehicles (3.14 vehicles) than did rural farm youths (2.59 vehicles). However, there was no significant difference between urban youths and rural farm youths in terms of the number of motor vehicles in the household (*p* > 0.05). 

#### 4.1.7. Neighbourhood Safety

The neighbourhood safety category includes four items, as outlined in Table 4. Overall, Chi-square analysis revealed a highly significant difference (*χ*^2^_1_ = 24.563, *p* < 0.001) between genders as to neighbourhood safety perception: 16.13% of males perceived that their environment was unsafe, whereas that percentage was more than double among females (39.33%). In terms of geographical location, Chi-square analysis showed a significant difference between participant perceptions (*χ*^2^_2_ = 19.815, *p* < 0.001). Rural desert youths perceived significantly safer neighbourhoods (90.22%), more so than urban youths (65.12%) and rural farm youths (69.00%). The differences between geographical locations were found when focusing on traffic questions. However, Chi-square analysis revealed no significant difference between geographical locations as to the crime rate questions (*p* > 0.05). Moreover, univariate ANOVA revealed that youths who perceived a higher crime rate in their neighbourhood were significantly (*p* < 0.001) less active (6195 steps per day) than those who perceived a low crime rate in their neighbourhood (8074 steps per day). Pearson’s correlation also revealed a significant positive relationship between neighbourhood crime rate and BMI (*r* = 0.106, *p* = 0.042), which means that youths who perceived a high crime rate in their neighbourhood had higher BMI. In addition, there were highly significant negative relationships between neighbourhood crime rate and PA (steps per day) (*r* = −0.281, *p* < 0.001).

#### 4.1.8. Access to Recreational Facilities

The only item on access to recreational facilities inquired whether “my neighbourhood has several free or low-cost recreation facilities, such as parks, walking trails, etc.”. As a whole sample, 61.06% of participants strongly disagreed or somewhat disagreed with this item. Chi-square analysis revealed a significant difference (*χ*^2^_1_ = 7.374, *p* = 0.005) between genders as to the perception of accessing recreational facilities. Among males, 70.43% perceived having poor access to recreational facilities, whereas the percentage of females was 56.74%. Regarding geographical locations, Chi-square analysis revealed a highly significant difference (*χ*^2^_2_ = 15.537, *p* = 0.005) between participants’ perceptions: 90.43% of rural desert youths reported that there was no access to recreational facilities in their neighbourhood, whereas the percentages of urban youths and rural farm youths were 74.30% and 89.22%, respectively. In addition, there was no significant difference between perceptions as to access to recreational facilities relative to PA level or weight status (*p* > 0.05).

The odds ratios (OR) and 95% confidence intervals (CI) of being overweight, having a high-risk waist circumference, and being inactive were applied to assess the association with built environmental variables, in order to identify whether neighbourhood perceptions predicted overweight status or inactivity. Youths who perceived an unsafe neighbourhood were 13% more likely to be overweight or obese (OR = 1.127, 95% CI = 1.003–1.266). Moreover, every household motor vehicle increased the risk of being overweight or obese by 17% (OR = 1.169, 95% CI = 1.004–1.360), as outlined in Table 4. 

Participants who perceived an unsafe neighbourhood were 34% more likely to be inactive—based on average steps per day (OR = 0.660, 95% CI = 0.508–0.857). The accessibility of places within easy walking distance of the home increases the likelihood of being active by 56%—based on average steps per day (OR = 1.560, 95% CI = 1.055–2.308), as outlined in Table 4 (CI: confidence interval).

#### 4.1.9. Perceived Benefits and Barriers

Students were asked to answer three additional questions. The first question was “how do you travel and go back from home to school”. In the second question, the students were asked to choose the most important three reasons for being active (motivation). The answer options were ‘built body/body image, weight loss, health, meet friends, recreational, competition, other reason’. Finally, in the third question, students were asked to choose the most common reasons for being inactive. The answer options were ‘No places to participate, not enough time, parents disagreement, social and traditional customs, peer pressure, weather, no company to practice with, health reason, laziness, expensive, not important, transportation and other’.

The reasons motivating being active were slightly diverse. In males, the most commonly reported reason for being active was health (25.93%), the second-most-common reason was built body/body image (20.11%), and the third was weight loss (17.99%); among females, the most commonly reported reason for being active was weight loss (45.25%), the second-most-common motivation was for health (24.58%), and the third was for recreational purposes and to fill free time (12.29%). The common reasons cited for being inactive, for both males and females, were having no places to participate (24.19% and 32.96%, respectively), not enough time (20.31% and 28.49%, respectively), and transportation difficulties (10.42% and 12.29%, respectively). These reasons were also common when the sample was divided by geographical location, private and public school, weight status, or PA level. Table 5 presents the number and percentage of youths travelling to school according to gender and geographical location. 

## 5. Discussion

The aim of this study was to evaluate the association between a neighbourhood’s environmental characteristics and health risk factors among KSA youth. The current study is one of the first targeting this population. The KSA has witnessed transformation and significant lifestyle changes during the last three decades, including digitalisation and diversification of various sectors. Thus, health risks, including hypertension, physical inactivity, sedentary behaviour, and the ever-increasing rate of obesity, have become prevalent in this society [38,55]. The present study shows that geographical areas impact PA and weight status. For instance, participants from the rural desert in this study were less active and had higher BMI compared to those living in rural farm or urban areas. This difference could possibly be due to several environmental factors, such as the climate and facilities which make it difficult for youths in rural desert locations to engage in regular PA, compared to their counterparts from other areas. The present findings support previous research reporting that neighbourhood parks are a community asset and can help provide opportunities for PA [56]. Furthermore, research has also reported that climate change, including heat waves, and the availability of neighbouring parks were related to lower/higher PA and sport participation [56,57,58]. On the contrary, participants from rural farmlands usually engage in active behaviour through completing farming-related activities that require physical movements, such as planting, ploughing, and harvesting. In addition, people residing in rural locations of the KSA do not consider PA or sport participation to be important for health due to cultural reasons. For instance, academic excellence is regarded as being of a higher status than engaging in PA, sport, or health-related interventions. Thus, some parents urge youths to take part in extra educational and spiritual classes rather than participate in leisure activities such as PA or sport. In addition to this, there is lack of available facilities, such as parks or sports grounds, and for both male and females, that makes it harder for youths to be physically active, especially girls. 

The current findings align with previous research reporting that cultural attitudes towards PA and environmental differences have been postulated as explanations for British participants being more active compared to their counterparts in Saudi Arabia [18]. Previous research also reported that people from the Middle East generally perceive more barriers to engaging in PA and report lower levels of PA compared to European countries [9]. It is apparent that the accessibility to PA and health and sport-related activities are easy for people residing in rural farm or urban environments in contrast to residence in the rural desert. Thus, the present findings support previous research outlining the fact that environments in which people consider PA participation to be vital for health are more likely to have residents who continue healthy and active lifestyles and face fewer barriers to engaging in PA, compared to places where PA participation is not a social norm [8]. 

The results of this study also revealed that some environmental variables such as residential density, neighbourhood safety, household motor vehicles, social environment, bicycling facilities, and accessibility to places were associated with PA or weight status. The results of this study also suggested that participants living in flats were more active compared to those living in houses. This might be because, in KSA, flat-style accommodations are located in high residential-density areas. Previous research highlighted the associations between residential density and both PA and weight status [59,60,61]. For example, previous research reported that high population density was negatively associated with BMI [61], while a large-sample study showed that BMI declined by 0.4 units for each 10,000 people/km^2^ increase in residential density [60]. Research also suggested that people were more likely to walk in high residential density areas [37]. High-population neighbourhoods enhance travel by walking and cycling [61]. Pedestrian activity can be promoted by more efficient provision of public transport, which reduces reliance on privately-owned transport. Increased levels of public-transportation-related activities driven by a corresponding reduction in owning/use of private transport can lead to a marked increase in day-to-day activity. Reflecting earlier work on urban form and travelling behaviour, studies found an association between environmental characteristics (such as population density and accessibility of nearby destinations) and measures of PA [54,62]. It has been suggested that many features of the built environment increase the use of public forms of transport while lowering the dependence on private transport. This in turn could increase PA while reducing body weight [62,63]. The number of motor vehicles in a household was associated with BMI in the current study. This result is supported by previous research suggesting that each additional hour spent per day in car increased the risk of being overweight or obese by 6% [63]. Conversely, each additional kilometre walked per day reduced the risk of being overweight or obese by 4.8%. 

The current findings revealed that 49% of youths who perceived that there were bicycling facilities in their neighbourhood were more likely to be active. The present findings align with previous research suggesting that participants who lived in neighbourhoods with general availability of sidewalk and cycling facilities were more active than their counterparts living in neighbourhoods with poor facilities [64]. These results were supported by other studies which found that the installation of infrastructure for cycles such as cycle lanes, as well as reductions in speed limits, led to a marked increase in daily bicycle use [65]. The current findings could be used by KSA policymakers and local professionals to encourage cycling and walking for health benefits. Previous research reported that integrating the promotion of walking and cycling into daily life could promote PA levels, and influence health, the environment, and the economy [66,67].

The current findings determined that neighbourhood safety (high-rate crime and traffic) was associated with PA and BMI. The measurement of safety is a key element of the physical environment that could influence PA behaviour. It is widely acknowledged that the perceptions of built environment of both youths and parents influence PA levels and weight status [28,68,69,70,71]. People’s perceptions of the safety of the surrounding built environment can vary depending on a range of variables, including perceived traffic density, the presence of pedestrian crossings, neighbourhood reputation, and the presence of graffiti and poorly lit areas. Previous research suggested that the perception of high crime levels and the total number of incidents of serious crimes were associated with a decreased MVPA engagement and increased BMI [72,73,74]. Furthermore, graffiti and vandalism were also associated with higher BMI [75].

The current findings suggested that partaking of PA was positively associated with social environment. For instance, youths who observed people engaging in PA in their neighbourhood were more active than those who perceived less activity in their social environment. It has been reported that friendly neighbourhoods with well-connected street networks have links with PA and improved health [76]. Previous research suggested that people residing in areas of high physical disorders such as graffiti, rubbish, and lack of green areas, as well as lower perceived safety, were less likely to encourage youths to use the local available playground [77]. Participants playing on their street was shown to be positively associated with parents perceiving the neighbourhood as a safe environment [78]. These findings support previous research claiming that constrained behaviour exhibited by parents has an inhibitive effect on their child’s levels of active transportation and PA levels in non-school hours [79].

The findings of the present study also revealed that accessibility to places such as shops, supermarkets, places, and transportation were associated with higher PA participation. Participants living in neighbourhoods with access to places were 56% more likely to walk further than their peers. Environmental factors, including availability of proximal non-residential destinations designed for pedestrian access, were correlated to PA. The current findings support previous research suggesting that people living in neighbourhoods with access to places, such as public transit, ease of walking, and going directly from sidewalk into store, were more likely to be active [1]. Thus, environmental factors such as access to local facilities and neighbourhood are vital and should be considered for PA and health-related interventions. For instance, it is imperative that youths have access to recreational venues in their neighbourhood, as this could help provide facilities to participate in PA [80].

When geographical areas of schools were considered, there were differences between participants’ perceptions from different geographical locations. More youths from the rural desert area lived in houses, had less accessibility in their neighbourhood, reported fewer aesthetic qualities, had more household motor vehicles, perceived less traffic, and lacked access to recreational facilities in their neighbourhood, compared to those from urban and rural farm areas. Moreover, comparing urban and rural farm groups, the results revealed that more urban youths lived in houses, had access to destinations, perceived their neighbourhood as having a good infrastructure, and reported more aesthetic qualities, less social environment, and more access to the recreational facilities compared to rural farm youth. 

No places to participate, not enough time, and lack of transportation were the most commonly reported reasons for being inactive for all participants across geographical areas. Moreover, there was some difference between motivations for being active among males and females. In males, the reason most commonly reported for being active was health; the second-most-common reason was built body/body image, and the third was weight loss. In females, the main reasons reported for being active were weight loss, health, and recreational aspects. The present findings support previous research conducted in KSA indicating that a lack of recreational and sports facilities was the most common barrier to PA, followed by lack of time and health conditions [81,82]. These findings were not surprising because of the limited access to facilities, especially for females, in which to join sport clubs, visit swimming pools, or engage in outdoor sports and recreation. 

## 6. Strength and Limitation

This study found its strength in evaluating the association between the neighbourhood environmental characteristics and health risk factors among KSA youth. To the best of the authors’ knowledge, this is the first study conducted in KSA focusing on neighbourhood environmental characteristics and health risk factors among youth. The additional strengths of this study were its unique feature of mixed-methods application of objective (pedometer) and subjective (questionnaires) tools, and the number of participants. This study contributes to the limited research on the subject and further highlights the necessity for future research and policy interventions, in order to improve the neighbourhood environmental characteristics concerning PA and the health risk factors for youths and the wider KSA population in different areas. The data in this study were collected from one region and the targeted population was youth. Collecting data across regions and different age groups may have generated different results. 

## 7. Conclusions

This study provided an important insight into the influence of neighbourhood environmental characteristics and health risk factors among KSA youths, including PA, weight status, and neighbourhood characteristics. This study also provides a base of knowledge for future research to build upon, design and deliver culturally tailored interventions to facilitate youths in the Al-Ahsa region of KSA. The present study also demonstrated the importance of accessible recreational facilities, including parks and sidewalks, especially in rural desert areas, where such amenities were reported to be lacking. Thus, health policies in the KSA could focus on developing the infrastructure to encourage regular PA engagement and BMI management among youth. This study also calls for future research to consider the climate change and tailored PA-, sport-, and health-related interventions according to the season of the year. For instance, during summer, activities conducted outdoors could be planned either early in the morning or post-sunset, or indoor PA- and sport-related activities could be arranged to remedy the low levels of PA among KSA youths, especially in Al-Ahsa region. Further research is needed to confirm the present findings, and this will therefore highlight the need for research and policy interventions to address the concerning neighbourhood environmental characteristics and health-risk factors among the youths of Al-Ahsa. This mixed-methods study provided the participants a voice and offered evidence to policy makers, public health departments, and local authorities for PA interventions based on the environment and community. To help better facilitate the needs of participants, future research could implement this approach as a framework in different geographical locations by considering the local environment and cultural and social factors, both within and outside of the KSA and the Al-Ahsa region.

## Figures and Tables

**Table 1 healthcare-12-01120-t001:** Mean *±* SD of the main dependent variables for the total sample and sub-samples.

Variable	Urban		Rural Farm		Rural Desert		Whole Group		Total *n* = 380
	Male *n* = 96	Female *n* = 86	Male *n* = 54	Female *n* = 49	Male *n* = 49	Female *n* = 46	Male *n* = 175	Female *n* = 160	
BMI	23.63 + 6.13	24.66 + 5.89	19.97 + 3.61	22.17 + 4.53	26.43 + 8.13	24.68 + 5.03	23.32 + 6.56	23.99 + 5.42	23.64 + 6.05
Abdominal Circumference	77.70 + 15.12	72.19 + 11.10	67.44 + 6.92	68.65 + 9.61	81.31 + 16.37	68.39 + 8.73	75.80 + 14.69	70.27 + 10.26	73.17 + 13.05
Total average daily Steps	8387 + 4106	5723 + 2682	10,281 + 4519	6440 + 2309	9546 + 6423	4362 + 2214	9180 + 4853	5580 + 2574	7460 + 4320

**Table 2 healthcare-12-01120-t002:** The subscales, with examples of items and methodology for the English version.

Scale/Subscale	Item Examples	Response Categories	Scoring
Residential density	What is the main type of housing in your neighbourhood?	Detached house/semidetached house/flat in small building/flat in big building/other	One item Range 1–5
Access to destination	It is within a 10–15-min walk to a transit stop (such as bus, train, trolley, or tram) from my home. Would you say that you…	4-point Likert: Strongly disagree/somewhat disagree/somewhat agree/strongly agree (SD to SA)	Mean of 3 items Range 3–12
Neighbourhood infrastructure	The sidewalks in my neighbourhood are well maintained (paved, with few cracks) and not obstructed. Would you say that you…	4-point Likert: SD to SA	Mean of 4 items Range 4–16
Aesthetic qualities	There are many interesting things to look at while walking in my neighbourhood. Would you say you…	4-point Likert: SD to SA	One item Range 1–4
Recreational facilities	My neighbourhood has several free or low-cost recreation facilities, such as parks, walking trails, bike paths, recreation centres, playgrounds, public swimming pools, etc. Would you say that you…	4-point Likert: SD to SA	One item Range 1–4
Social environment	I see many people being physically active in my neighbourhood doing things like walking, jogging, cycling, or playing sports and active games. Would you say that you…	4-point Likert: SD to SA	One item Range 1–4
Household motor vehicles	How many motor vehicles in working order (e.g., cars, trucks, motorcycles) are there for your household?	Number	Number of household motor vehicles
Neighbourhood safety	The crime rate in my neighbourhood makes it unsafe to go on walks at night. Would you say that you…	4-point Likert: SD to SA	Mean of 4 items Range 4–16

**Table 3 healthcare-12-01120-t003:** The correlation coefficients of weight status and PA with selected environmental variables.

		Safety	Crime	Traffic	Access	Infrastructure	Recreation Facilities	Social Environment	Aesthetics
BMI	R value	0.047	0.106 *	−0.054	0.025	0.003	−0.031	0.008	−0.001
	*p* value	0.368	0.042	0.302	0.646	0.952	0.550	0.870	0.989
Waist circumference	R value	−0.051	−0.046	−0.012	0.033	−0.012	−0.053	0.101	0.076
	*p* value	0.327	0.372	0.818	0.545	0.837	0.314	0.052	0.143
Steps	R value	−0.235 **	−0.281 **	−0.045	−0.018	−0.076	−0.068	0.077	0.009
	*p* value	0.000	0.000	0.415	0.763	0.204	0.223	0.164	0.868

** Correlation is significant at the 0.01 level. * Correlation is significant at the 0.05 level.

**Table 4 healthcare-12-01120-t004:** The odds ratios for environmental variables and likelihood of subjects being overweight/obese or meeting the recommended physical activity level.

	Normal/Overweight			Inactive/Active (Steps per Day)		
Items/Category	Odds Ratios	95% CI	*p* Values	Odds Ratios	95% CI	*p* Values
Neighbourhood safety	1.127	1.003–1.266	0.045	0.660	0.508–0.857	0.002
Household motor vehicles	1.169	1.004–1.360	0.004	1		
Social environment	1					
Bicycling facilities				1		
Accessibility to places				1.560	1.055–2.308	0.047

**Table 5 healthcare-12-01120-t005:** Travelling to school according to gender and geographical location.

	Urban				Rural Farm				Rural Desert				Whole Group					
Travelling Type	Male *n* = 96		Female *n* = 86		Male *n* = 54		Female *n* = 49		Male *n* = 49		Female *n* = 46		Male *n* = 199		Female *n* = 181		Total *n* = 380	
	*n*	%	*n*	%	*n*	%	*n*	%	*n*	%	*n*	%	*n*	%	*n*	%	*n*	%
Walking	3	3.12	2	2.33	32	59.26	14	28.57	2	4.08	3	6.52	37	18.59	17	9.39	54	14.21
Cycling	0	0	0	0	0	0	0	0	0	0	0	0	0	0	0	0	0	0
Motor vehicles	93	96.88	84	97.67	22	40.74	35	71.43	47	95.92	43	93.48	162	81.41	164	90.61	326	85.79

## Data Availability

Data are contained within the article.

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
