# Peer review of "Association between Young People’s Neighbourhoods’ Characteristics and Health Risk Factors in Saudi Arabia"

_healthcare, 2024, doi:10.3390/healthcare12111120_

Round 1
Reviewer 1 Report
Comments and Suggestions for Authors
Brief Summary:
This manuscript explores the association between neighborhood Characteristics and Health Risk Factors for youth in Saudi Arabia by performing group comparison on 335 secondary-school students. This study sheds light on the significant statistical difference in physical activity levels and overall health among youths residing in various neighborhoods. Such findings are helpful for researchers and policymakers aiming to enhance neighborhood environments to foster healthier youth lifestyles in Saudi Arabia.
However, I have several concerns that require clarification:
- Incomplete Data Reporting: For most discussions in Section 5, It does not have detailed data and statistics results. It is crucial to add comprehensive data and analytical reports to support your statements.
- Ambiguity in Section 5.5: The discussion on the relationship between PA and the social environment raises questions.
- Although authors define PA as daily steps in the abstract, its presentation within the text remains unclear. A clear restatement of the PA definition in this section is recommended.
- You mentioned the correlation between steps per day and social environment is 0.137 with p-value 0.008. However, the two numbers inside the Table 3 is 0.077 and 0.164
- Similarly, the statement about the two objects’ correlation in abstract is not supported by Tabel 3.
- The Authors explore correlations between BMI/Waist Circumference/Steps and various environmental factors, potentially overlooking the internal correlation of these factors with the participants' residential areas. An intra-group correlation analysis might elucidate whether different strategies should be employed across diverse areas.
- For section 4, you may need to add more analysis in comparing the dependent variables between different areas.
Minor issues:
1. Typo: “P<05” on row 204 at page 5
2. Clarification is needed on how the “Total daily Steps/week” in Table 1 was calculated. Is it an average of daily steps over a single week or multiple weeks. If it's the latter case, the duration of the observation period should be noted.
3. A percent sign % is missing on row 216 at page 5 inside the bracket (82.52)
Author Response
Please find attached the responses to reviewer 1 comments

Reviewer 2 Report
Comments and Suggestions for Authors
Thank you for considering the Journal of Healthcare for your paper. The paper needs to address some main issues to be suitable for publication. The overall structure of the paper is well-organised. There is a need to proofread the paper, as there are some issues with the paper flow. Should the spelling follow USA or UK English? "Neighborhood" or just "neighbourhood"?
The authors claimed that this is the first time this kind of consideration is done in an Arabian context. It is a strong statement and needs to be justified in more detail. How do the authors know it? The paper seems to use the same neighbourhood characteristics that have been used in Western literature. From my understanding, these can be different in different geographical locations and based on different cultures. It is expected that authors add to the list of characteristics based on their study in this specific culture and geographical location. Would climate be effective? Having hot and dry summers might also affect the results. Girls might have more limitations to go out freely and engage in physical activity compared to Western countries. This should be acknowledged, especially when authors talk about social environment, safety, and barriers. What do you mean by "English version"? What would be the difference between English and Arabian?
The discussion talks about children and youth. They should be separated. On page 9, line 376, comparing studies about adolescents and children in the discussion makes it difficult to understand the purpose of this research. These age groups are different from youth. Please refer to your results for studies on youth only (there is a large number of literature on this topic and youth participants in the literature).
The implementation needs to be detailed. What can we do with the findings of this research? How would it be helpful for me as a reader from another geographical location? The conclusion is not written as a proper conclusion. What can future researchers do based on this research? How?
I am assuming that on Page 7, line 281, it should be Table 2, not 4.
Addressing these concerns will enhance the paper's quality and increase its suitability for publication.
Author Response
Please find attached responses to reviewer 2 comments

Round 2
Reviewer 1 Report
Comments and Suggestions for Authors
I appreciate the authors' efforts on the revision and all my comments have been addressed. I believe the manuscript is in good quality now.
Author Response
Dear reviewer 1,
Thank you. Please see attached.

Reviewer 2 Report
Comments and Suggestions for Authors
Thank you for addressing the comments. My only comment is that the authors need to add in the limitations or conclusion a discussion on the role of climate on PAs and provide direction on how it can be considered in future research. Additionally, the authors still need to add an implementation section to explain how this paper can be used as a framework for future research, as claimed in the conclusion and cover letter. My question of "So what can we do with these findings?" remains, as it is not clear how this framework can be applied to different geographies. Also, what are the takeaways for readers from different locations?
Author Response
Dear reviewer 2,
Please see attached.
